# Fluorescence Super-Resolution Imaging Chip for Gene Silencing Exosomes

**DOI:** 10.3390/s24010173

**Published:** 2023-12-28

**Authors:** Gaoqiang Yin, Tongsheng Qi, Jinxiu Wei, Tingyu Wang, Zhuyuan Wang, Yiping Cui, Shenfei Zong

**Affiliations:** Advanced Photonics Center, School of Electronic Science & Engineering, Southeast University, Nanjing 210096, China; 220211711@seu.edu.cn (G.Y.); 213162551@seu.edu.cn (T.Q.); 230208633@seu.edu.cn (J.W.); 230208632@seu.edu.cn (T.W.); wangzy@seu.edu.cn (Z.W.); cyp@seu.edu.cn (Y.C.)

**Keywords:** exosomes, immunotherapy, PD-L1, microfluidic chip, DNA-PAINT, integrated platform

## Abstract

Tumor cell-derived extracellular vesicles and their cargo of bioactive substances have gradually been recognized as novel biomarkers for cancer diagnosis. Meanwhile, the PD-L1 (Programmed Death-Ligand 1) protein, as an immune checkpoint molecule, is highly expressed on certain tumor cells and holds significant potential in immune therapy. In comparison to PD-L1 monoclonal antibodies, the inhibitory effect of PD-L1 siRNA (small interfering RNA) is more advantageous. In this article, we introduced a microfluidic chip integrating cell cultivation and exosome detection modules, which were intended for the investigation of the gene silencing effect of PD-L1 siRNA. Basically, cells were first cultured with PD-L1 siRNA in the chip. Then, the secreted exosomes were detected via super-resolution imaging, to validate the inhibitory effect of siRNA on PD-L1 expression. To be specific, a “sandwich” immunological structure was employed to detect exosomes secreted from HeLa cells. Immunofluorescence staining and DNA-PAINT (DNA Point Accumulation for Imaging in Nanoscale Topography) techniques were utilized to quantitatively analyze the PD-L1 proteins on HeLa exosomes, which enabled precise structural and content analysis of the exosomes. Compared with other existing PD-L1 detection methods, the advantages of our work include, first, the integration of microfluidic chips greatly simplifying the cell culture, gene silencing, and PD-L1 detection procedures. Second, the utilization of DNA-PAINT can provide an ultra-high spatial resolution, which is beneficial for exosomes due to their small sizes. Third, qPAINT could allow quantitative detection of PD-L1 with better precision. Hence, the combination of the microfluidic chip with DNA-PAINT could provide a more powerful integrated platform for the study of PD-L1-related tumor immunotherapy.

## 1. Introduction

As an immune checkpoint molecule, the PD-L1 (Programmed Death-Ligand-1) protein plays a crucial role in the human immune system. It possesses critical immunological functions and can regulate the immune response of T cells toward tumor cells, thereby avoiding excessive damage to surrounding tissues [1,2]. When PD-L1 binds with its receptor PD-1 on T cells, it inhibits the immune function and prevents the immune system from attacking normal tissues. However, in the tumor environment, most tumor cells utilize this mechanism to evade immune system surveillance. Tumor cells mimic normal cells and highly express PD-L1, making it difficult for the immune system to recognize and attack tumor cells. Therefore, blocking or reducing the expression of PD-L1 in tumor cells is of great significance [3]. PD-L1 monoclonal antibody is usually used in PD-L1-based immunotherapy. It can bind to PD-L1 on tumor cells and avoid the interaction of PD-L1 and PD-1 on T cells, thereby activating T cells and enhancing the immune system’s ability to attack tumors. For example, Durvalumab is a kind of PD-L1 monoclonal antibody belonging to the IgG1 class, which has been approved by the FDA and can be used to treat UC (Urothelial Carcinoma) and NSCLC (Non-Small Cell Lung Carcinoma). Inhibiting PD-L1 has been proven to be a viable treatment strategy and has achieved significant success [4]. However, during treatment, continuous delivery of PD-L1 monoclonal antibodies is required to maintain the inhibitory effect. The therapeutic efficacy also varies significantly among different patients [5]. Compared to PD-L1 monoclonal antibodies, PD-L1 siRNA (small interfering RNA) has a distinct advantage, as it can generate benefits in a more direct manner. PD-L1 siRNA utilizes gene silencing mechanisms to interfere with the transcription and translation processes of the target gene, effectively reducing the synthesis of PD-L1 protein from its source. Additionally, PD-L1 siRNA can sustainably impact the expression of the target gene, thereby achieving long-lasting inhibitory effects [6,7,8,9]. Moreover, the dosing interval is 2 years for each administration [8]. PD-L1 siRNA can be designed with specific sequences to precisely inhibit the expression of target genes, which means personalized treatment can be formulated based on the genetic characteristics and conditions of patients, thus improving treatment efficacy.

Exosomes are cell-derived vesicles that can be released into various body fluids, such as blood and saliva. They carry a variety of proteins, including PD-L1 [10,11,12,13]. Compared to the traditional direct detection of tumor cells, extracellular vesicle (EV) detection is a non-invasive method that does not require tissue biopsy or cell sampling [14]. In addition, exosomes are widely distributed in the circulatory system, dynamic systemic detection can be achieved by regularly collecting and analyzing exosome samples. The convenient operation and analysis process of EV testing has shown tremendous potential as a non-invasive biopsy method [15,16]. In this study, the expression of PD-L1 on exosomes was chosen to validate the gene-silencing effect of siRNA.

Although exosome detection has many advantages compared to direct tumor cell testing, the size of exosomes is much smaller compared to tumor cells [17]. The small size and heterogeneity of exosomes make quantification and accurate estimation of exosomes still challenging. 

Microfluidic chips are small-sized chips that manipulate fluids in microscale spaces, which can achieve the processing and analysis of samples with small volumes [18,19]. The increased surface area of the channels provides more reaction interfaces, improving contact and reaction efficiency between reagents. Therefore, microfluidic chips have unique advantages in realizing rapid analysis and efficient processing.

DNA-PAINT (DNA point accumulation in nanoscale topography) is an important subtype of single-molecule localization microscopy (SMLM), which was proposed by Ralf Jungmann and colleagues in 2014 [18]. DNA-PAINT achieves multi-color super-resolution fluorescence imaging by exploiting the complementary nature of DNA strands. In this technique, the highly specific binding between double-stranded DNA leads to the spontaneous binding of two short, complementary DNA strands in solution. By attaching one of the DNA strands to the target molecule as a docking strand, and attaching the complementary strand to a fluorophore as an imaging strand, the imaging and docking strands can undergo reversible binding and dissociation. DNA-PAINT utilizes a total internal reflection (TIRF) illumination scheme. When the imaging strand (modified with a fluorescent molecule) binds to the docking strand, the fluorescent molecules are excited, producing a fluorescence on a signal. Due to the limited illumination depth of TIRF, a fluorescence off signal would be produced when the imaging strand dissociates. This on-off “blinking” effect allows precise localization of the targets with a spatial resolution down to 10–20 nm [20]. DNA-PAINT offers some advantages compared to other super-resolution imaging methods. First, the imaging rate can be controlled by adjusting the length of DNA strands and the composition of the imaging buffer. Second, this technique effectively avoids the issue of photobleaching. Third, DNA-PAINT allows for the attachment of any fluorescent dye to the imaging strand, enabling multicolor imaging (e.g., Exchange-PAINT) without a “fluorescent ceiling” [19,20,21]. Ralf Jungmann et al. further explored the binding process of DNA strands and successfully proposed an innovative technique called qPAINT. Through detailed analysis using qPAINT, precise quantification of nanoscale substances, such as proteins and DNA, can be achieved. qPAINT is easy to operate and highly stable, opening up new avenues for quantitative studies of nanoscale materials. Therefore, in this study, DNA-PAINT and qPAINT were employed to investigate spatial localization and quantitatively analyze PD-L1 proteins on exosomes of HeLa cells, aiming to explore the inhibitory effect of siRNA on PD-L1 protein in tumor cells. 

The novelty of this study lies in the combination of a microfluidic chip and super-resolution imaging techniques, proposing a microfluidic chip that integrates cell culture and exosome detection. This study provides a more powerful approach for the study of exosome-based cancer immunotherapy.

## 2. Experimental Section

### 2.1. Materials

Polyethyleneimine (PEI, 408 727) was purchased from (Nanjing, China) Kaiji Biotechnology Co., Ltd. Ultraltration tube (50 kDa, UFC5050), beta-mercaptoethanol, glucose, glucose oxidase, and hydrogen peroxide oxidase were purchased from (St. Louis, MO, USA) Sigma-Aldrich. Exosome-depleted fetal bovine serum (Exo-FBS) was purchased from (Palo Alto, CA, USA) System Bioscience (SBI). Glutaraldehyde (GA, 50% concentration) was purchased from (Shanghai, China) MACKLIN Company. Sodium borohydride (NaBH_4_) was purchased from Aladdin Bio-Chem Technology (Shanghai, China) Co., Ltd. Alexa Fluor 647@rabbit anti-PD-L1 (bs-10159R-AF647) and phosphate-buffered saline (PBS, 10 mM, pH 7.4) were purchased from Beijing Biosynthesis Biotechnology (Beijing, China) Co., Ltd. TN buffer (50 mM Tris, 10 mM NaCl, pH 8.0) was purchased from Runode Biotech (Suzhou, China) Co., Ltd. Human cervical cancer cells (HeLa) were purchased from Stem Cell Bank, Chinese Academy of Sciences (Beijing, China). Bovine serum albumin (BSA) was purchased from (St. Louis, MO, USA) Sigma-Aldrich. Normal goat serum (NGS) was purchased from (Wuhan, China) Poisedon Bioengineering Co., Ltd. DBCO-sulfo-NHS ester was purchased from (Jena, Germany) Jena Bioscience. RNA TransMate transfection reagent, CD63 adapter, PD-L1 adapter, PD-L1 siRNA, PD-L1 docking chain, and PD-L1 imaging chain were synthesized by Sangon Biotech (Shanghai, China) Co., Ltd. The sequences are shown in Appendix A. 

### 2.2. Exosome Extraction

Cells were cultured under standard cell culture conditions (5% CO_2_, 37 °C). When the cell confluence reached 70%, 10 mL of cell culture medium was taken from the culture flask and centrifuged at 4 °C to remove cells and cell debris from the medium (3000× *g*, 15 min). Subsequently, the supernatant was collected and centrifuged again at 4 °C (10,000× *g*, 45 min) to remove large vesicles presented in the supernatant. The supernatant was then subjected to additional centrifugation at 4 °C using an ultracentrifuge (150,000× *g*, 70 min, twice). A small amount of PBS solution was added to resuspend the exosomes, which were then aliquoted and stored at −72 °C for future use.

### 2.3. Gene Silencing

Cells were seeded in an eight-well chambered slide and incubated in a CO_2_ incubator for 24 h. Next, the siRNA/RNATransMate complex was prepared as follows. Two centrifuge tubes were prepared, followed by the addition of 190 μL (marked as tube A) and 200 μL (marked as tube B) antibiotic-free and serum-free cell culture medium, respectively. Then, they were shaken gently. Next, 12 μL of RNATransMate was added to Tube A, and 2 μL of siRNA stock solution (40 pM) was added to Tube B. Next, Tube A and Tube B were mixed together and kept still for 5 to 10 min.

The cell culture medium was removed from the eight-well chambered slide. The chambers were washed three times with PBS. The prepared siRNA/RNATransMate complex and complete culture medium were added to the eight-well chambered slide in proportion. The concentration of siRNA in the culture medium was 30 nM, 20 nM, 10 nM, and 5 nM, respectively. The eight-well chambered slide was placed back into the CO_2_ incubator and incubated for 24 h. A confocal microscope was used to observe the uptake of siRNA by cells at 6 h, 12 h, and 24 h during the RNA transfection process.

### 2.4. Immunofluorescence Staining

After transfection, the transfected tumor cells were cultured in standard growth conditions until the cell confluence reached 70–90%. The cells were then removed from the cell culture incubator. The cell culture medium was aspirated, and each chambered slide of the eight-well chambered slide was washed with PBS solution three times. Next, 400 μL of fixation solution (4% paraformaldehyde (5 mL) and 50% glutaraldehyde (10 μL)) was added. After 20 min, the chambered slides were washed with PBS solution three times. Then, 400 μL of NaBH_4_ solution (1 mg/mL) was added, and the eight-well chambered slide was shaken for 7 min to reduce the aldehyde groups in the fixation solution.

Next, approximately 200 μL of NGS was added and kept still for 90 min to block non-specific binding sites. Then, the wells were washed with PBS solution three times. Finally, Alexa Fluor 647-conjugated rabbit anti-PD-L1 antibody (1 μL, 1 mg/mL) and PBS solution (400 μL) were added to react for 12 h at 4 °C in the dark. After the reaction, the chambered slides were washed with PBS solution three times. Finally, a small amount of PBS solution was added. Thus, the immunofluorescent staining of the tumor cells was completed and the cells were ready for subsequent imaging procedures.

### 2.5. Preparation of PDMS Microfluidic Chips

The cover glass was placed into a freshly prepared Piranha solution (a mixture of 98% concentrated sulfuric acid and 30% hydrogen peroxide with a volume ratio of 3:1). After sonicating for 1 h, the cover glass was rinsed thoroughly with deionized water and dried with argon gas. Polydimethylsiloxane (PDMS) was mixed with the curing agent at a ratio of 10:1. The mixture was poured into a resin-positive mold made by a 3D printer. Prior to use, the resin-positive mold should be sonicated in a fluoride solution for 5 min. Due to the potential deformation of the resin at high temperatures, the PDMS-filled resin-positive mold was placed in a 60 °C oven for 2 h to cure the PDMS. Then, holes were drilled in the PDMS chip, which was further cleaned with alcohol using ultrasonication and dried for later use. Lastly, the cleaned PDMS chip and cover glass were treated with oxygen plasma for 5 min (oxygen flow rate of 100 sccm, alternating current of 50 kHz, 100 W). Then, the cover glass was pressed gently onto the PDMS chip to create an integrated PDMS chip for cell culture, exosome capture, and detection.

### 2.6. Cultivation and Transfection of Cells

The culture medium was removed from the cell culture dish and 200 μL of trypsin solution (0.25%) was added for digestion for 1 min. Then, the trypsin solution was aspirated, and 500 μL of the corresponding complete culture medium was added. The adhered tumor cells were detached gently from the dish. The tumor cells along with the culture medium were transferred to a microfluidic chip that was sterilized with ultraviolet light. Then, the chip was placed back into the cell culture incubator to allow reattaching of the tumor cells.

When the cell confluence reached 60–90%, the culture medium was removed from the microfluidic chip, and the chip was washed three times with PBS. Then, 50 μL of culture medium containing different concentrations of PD-L1 siRNA (40 nM, 30 nM, and 20 nM) was added. A blank control group was set as well. After 24 h, the culture medium was removed and the chip was washed three times with PBS to remove excess siRNA. Finally, 50 μL of siRNA-free complete culture medium was added and the cells were allowed to grow for 3 to 5 days to secrete a sufficient amount of exosomes for subsequent capture and detection experiments.

### 2.7. Specific Capture and Detection of Exosomes

Aldehyde functionalization of the substrate: First, 15 μL of a 0.5% (*w*/*v*) PEI solution was injected into the channel at a rate of 0.5 μL/min for 5–10 min. The solution was simultaneously extracted through the outlet at the same rate. Then, 15 μL of PBS solution was injected into the channel at a rate of 3 μL/min to wash the excess PEI solution. Next, 15 μL of a 2.5% (*w*/*v*) GA solution was injected into the channel at a rate of 0.5 μL/min. GA would react with the amino groups on the surface of PEI modified glass slide. After 30 min, the channel was washed with PBS solution.

Biofunctionalization of substrates: 15 μL of a 5 μM CD63 aptamer solution was injected into the channel at a rate of 0.5 μL/min for 10 min. The aptamer was modified with an amino group at the 3’ end, allowing it to react with GA on the glass surface inside the channel. Then the channel was washed with PBS solution. To prevent non-specific binding of exosomes to the glass surface, 15 μL of a 2% (*w*/*v*) BSA solution was injected into the channel at a rate of 0.5 μL/min for 30 min at room temperature to block non-specific binding sites. Finally, the channel was washed three times with PBS solution.

Exosome Capture: The culture medium containing exosomes was extracted into the channel at a rate of 0.5 μL/min using an infusion pump. After extraction, the medium was kept still in the chip for 10 min, allowing capture of the exosomes via CD63 aptamers. Then, the excess culture medium was washed away. To further improve reliability, dual-color fluorescence co-localization was used to locate and detect the exosomal PD-L1 protein. Next, 10 μL of a 10 nM PKH26 membrane dye solution was injected into the channel at a rate of 0.5 μL/min to stain the membrane of the captured exosomes, the excess PKH26 dyes were also washed away with PBS. Finally, 30 μL of a 1 nM Alexa Fluor 647-conjugated PD-L1 aptamer solution was injected into the channel at a rate of 0.5 μL/min. This aptamer can specifically bind to the PD-L1 protein on the surface of exosomes and label the PD-L1 protein. 

### 2.8. DNA-PAINT Imaging and qPAINT Analyzing of the Exosomal PD-L1

DNA-PAINT imaging: After washing the channel with PBS, due to the small volume of the channel, we introduced high-concentration imaging chains to obtain better scintillation images. First, 10 μL of imaging buffer containing DNA imaging chains (50 nM) was introduced into the channel for super-resolution fluorescence imaging. The microscope was set to EPI and TIRF modes. The sample was put on the stage for 20 min to eliminate the drift error. During the imaging process, we first determined the location of exosomes in the PKH26 channel, then turned on the 642 nm laser and switched to the Cy5 channel for DNA-PAINT imaging. In the fluorescence acquisition process, the exposure time was 100 ms, and 5000 frames of images were collected for the reconstruction of super-resolution images.

qPAINT analysis of the exosomal PD-L1: Firstly, the photoswitching curve of the DNA strand was binarized, then a suitable threshold was set as the standard to judge the dark state or light state. The bright state was considered as a combination of the imaging strand and the DNA docking strand. The interval between two adjacent dark regions was defined as the dark duration *τ_d_*. The scintillation curve was fitted to obtain the corresponding *τ_d_* distribution diagram of an exponential function. According to Formula (1), the value of $\tau_{d}^{*}$ can be approximately considered as the value of t when the value of function f(t) is 1 × 10^−1^.
(1)$$f(t)=1-exp(-t/\tau_{d}^{*})$$

Next, the content of PD-L1 proteins on the surface of exosomes can be obtained according to the number of binding sites represented by N in Formula (2)
(2)$$N=(\xi *\tau_{d}^{*}{)}^{-1}$$
where ξ is the inflow rate of a given imaging chain, the value of ξ is $k_{on}\times c_{i}$. $k_{on}$ is the secondary binding rate of DNA sequence, and $c_{i}$ is the concentration of the imaging chain. For the same reaction conditions, the value of ξ is constant, so it can be concluded that the number of binding sites N is proportional to ($\tau_{d}^{*}$)^−1^. 

The super-resolution microscope was used to obtain 5000 frames of scintillation images (Cy5 channel). The original image data were imported into Picasso software 1.0. The reconstructed exosome image was selected, and the analysis window of Picasso software was opened to obtain $\tau_{d}^{*}$. Then the calculated constant ξ was input to obtain the corresponding exosomal PD-L1.

### 2.9. Instruments

SMLM and DNA-PAINT images were acquired using a Zeiss Elyra P.1 super-resolution microscope equipped with 405 nm (100 mW) laser, 561 nm (100 mW) laser, 642 nm (100 mW) laser, 100×/1.46 oil immersion objective, and an Andor EM-CCD camera (iXon DU897). TIRF illumination was used for wide-field imaging in the experiment. A Longer Pump LSP04-1A four-channel syringe pump was used, the rated linear thrust was greater than 180 N, and the syringe specification was set to 2 mL. The pump was used to inject the reaction reagents into the microfluidic channel at the required time and speed. The nanoArch P150 3D printer manufactured by Mofang Precision Company was used to print photosensitive resin, which was equipped with a UVLED light source (405 nm). The exposure time was 4 s, and the optical accuracy was 25 μm.

## 3. Results and Discussion

### 3.1. The Principle of the Inhibition of PD-L1 Expression in Tumor Cells by PD-L1 siRNA

Currently, the use of PD-1 or anti-PD-L1 monoclonal antibodies to inhibit the PD-1/PD-L1 interaction has been proven effective in the treatment of several cancers [22]. However, many patients do not achieve the expected results with this treatment [3]. Therefore, our attention has turned to small interfering RNA (siRNA), which could be used as an alternative method to block immune checkpoints. siRNA is a substitute approach that works by reducing the expression of PD-1 or PD-L1 to block their interaction. PD-1 siRNA and PD-L1 siRNA downregulate the expression of PD-1 or PD-L1 proteins by blocking the transcription process of PD-1 or PD-L1 mRNA within cells. 

We developed a microfluidic chip capable of cell culture and exosome detection. Tumor cells were transferred into the chip for gene-silencing experiments. Additionally, the chip facilitated the capture of cancer cell-secreted exosomes. Finally, the inhibition of PD-L1 protein expression on the surface of tumor cell-derived exosomes by siRNA was validated using immunofluorescence assay. The photo of the microfluidic chip is shown in Appendix A. The disc-shaped part of the chip is used for cell culture, making it convenient to directly secrete the required exosomes. The straight passages are used for exosome detection. The part between the disc-shaped part and the straight passage is designed to be tortuous, which can prevent the cells or large vesicles from entering the detection area with the assistance of an extremely slow flow rate.

The structure of the microfluidic chip is shown in Figure 1. The circular part is the cell culture area. The cell culture media are introduced into the straight detection unit through the curved channel. During the detection process, the CD63 aptamers are first modified on the inner wall of the straight channel using PEI and GA to capture exosomes. CD63 is known as the lysosomal-associated membrane protein 3 (LAMP3), a member of the four-pass transmembrane protein superfamily, and is widely expressed on the surface of many cell types [23,24,25]. The abundance of CD63 on the surface of exosomes is typically higher than other biomarkers (such as CD9 or CD81), making it easier and more efficient to capture exosomes using CD63 aptamers [26]. So, CD63 is a universal protein in exosomes and usually is used as the capture point [27].

When the exosomes were captured, a fluorescently labeled PD-L1 aptamer and PKH26 membrane dye solution were introduced into the straight channel to label the PD-L1 protein and membrane of the exosomes. Subsequently, the exosomes in the channel were imaged.

PKH26 is a kind of fluorescent dye that can stably bind to the lipid bilayer membranes of the exosomes. Alexa Fluor 647 (AF647) was modified on the PD-L1 aptamer. For a specific reaction site, the binding of exosome and PD-L1 aptamers would induce a perfect overlap of AF647 and PKH26 channels in the images. The pseudo colors of PKH 26 and AF647 channels are set as red and green, respectively. So, the localization of exosomes using two-color fluorescence co-localization of PKH26 and AF647 would be observed as yellow sites. The open-source Image-Pro Plus software 6.0 was used for co-localization analysis. More details could be found in our previous work on eliminating nonspecific bind sites using fluorescence colocalization [28,29].

The capture probe (CD63 aptamer) and recognition probe (PD-L1 aptamer-AF647) have high binding affinity to the CD63 protein and PD-L1 protein on the surface of exosomes. It is difficult for the probes to diffuse or detach from the exosomes without specific dissociation steps [27]. 

#### 3.1.1. Uptake of siRNA by Tumor Cells

We first studied the uptake of siRNA by tumor cells. To verify the gene-silencing effect of PD-L1 siRNA on tumor cells, immunofluorescence staining was used to quantitatively analyze the expression levels of PD-L1 protein on the surface of normal and transfected cells. HeLa cells were co-cultured with PD-L1 siRNA at a concentration of 30 nM for 24 h. The cells were then subjected to immunofluorescence staining and observed under a confocal microscope. The results are shown in Figure 2, where the red fluorescence represents AF647 on the PD-L1 antibody, indicating that the antibody has successfully connected to the PD-L1 protein on the tumor cell membrane surface through the antigen-antibody reaction. Figure 2A,C show the comparison of the expression levels of the PD-L1 protein on the surface of HeLa cells before and after transfection.

By comparing the fluorescence intensity of PD-L1 antibody in the figure, we can see that the fluorescence intensity of untreated cells is significantly higher than that of transfected cells, indicating that PD-L1 siRNA has successfully entered the cells and interacted with mRNA, resulting in the significant reduction of PD-L1 expression. To better show the inhibition effect of the siRNA, fluorescence intensities were calculated by selecting regions of interest (as shown by the red boxes) using the measurement function of the FV1000 software 4.2 equipped with the confocal microscope. The results obtained by averaging multiple groups of data are shown in Figure 2E. It can be seen that the fluorescence intensity of PD-L1 protein on the surface of transfected cells is obviously lower than that of untreated cells.

#### 3.1.2. Capture and Detection of Exosomes Secreted by Cells

After completing cell gene silencing in the chip, the cell culture medium was introduced into the capture channel. The CD63 aptamers modified at the bottom of the channel were used to capture exosomes. The membrane of captured exosomes was labeled by PKH26. The PD-L1 aptamer modified with AF647 was used to label PD-L1 on the surface of exosomes. The location of exosomes was determined by dual-color fluorescence colocalization of PKH26 and AF647. 

Figure 3 shows the two-color fluorescence colocalization image of exosomes captured in the microfluidic chip. The red channel is exosome membranes stained with PKH26. The middle channel is PD-L1 proteins on exosomes labeled by AF647 aptamers, the right figure is the merged image. Figure 3D–F is an enlarged view of the boxed areas in Figure 3A–C. From the figure, it can be clearly seen that the colocalization effect of PKH6 and AF647 is excellent, which proves that the microfluidic channel can successfully capture exosomes. This provides an experimental basis for the subsequent quantitative analysis of exosomal PD-L1 proteins.

To investigate the specificity of the aptamer probe toward PD-L1, we conducted a control experiment using non-specific probes. The experimental results are provided in Appendix A. As can be seen, a negligible signal corresponding to the probes (642 channel) was detected, resulting in poor colocalization between the probes and the membrane dye (543 channel, i.e., exosomes) channels. Hence, the nonspecific probes cannot bind to PD-L1 on the exosomes. This can prove that the probes used in the article hold good specificity toward exosomal PD-L1.

We utilized a fluorescence colocalization strategy to guarantee the specificity of the results. Only those spots with both the signals of exosomes (PKH26) and PD-L1 (Alexa Fluor 647) were taken as the specific sites, which ensured that the objects we investigated were indeed PD-L1 on exosomes. Those spots with only the signals of Alexa Fluor 647 were nonspecific binding sites and were abandoned in subsequent analyzing procedures. Hence, our method holds excellent specificity. We have proposed and utilized such a colocalization strategy in several of our previous works, all focusing on exosome imaging or detection [28,29,30].

Subsequently, the contents of PD-L1 proteins on exosomes transfected with different concentrations of PD-L1 siRNA were compared by measuring the fluorescence intensities of AF647. The results are shown in Figure 4. With the increase of PD-L1 siRNA concentration, the fluorescence intensity of PD-L1 protein on the surface of HeLa cell exosomes gradually weakened, indicating that more siRNA has entered the cell and bound to mRNA related to PD-L1 transcription, preventing the expression of PD-L1.

### 3.2. Quantitative Analysis of Exosomal PD-L1 by DNA-PAINT

Next, we continued to quantitatively analyze PD-L1 on exosomes using DNA-PAINT. Similarly, HeLa cells were cultured in the integrated chip and gene silencing was performed using PD-L1 siRNA. After that, the exosomes were captured in the microfluidic channel. The captured exosomes were observed by a super-resolution microscope, and the SMLM images were obtained. The PD-L1 protein on exosomes was quantitatively analyzed by qPAINT. The experimental principle is shown in Figure 5.

First, DBCO-sulfo-NHS ester was used to couple the antibody targeting PD-L1 protein with the docking chain to form specific labeled probes, which could bind to the PD-L1 protein on the surface of exosomes captured in the microfluidic channel. Under the TIRF illumination mode, the imaging chain would emit fluorescence when it binds to the docking chain. When the imaging chain detaches and moves away from the docking chain, the fluorescence disappears due to limited illumination depth. So, the fluorescence scintillation phenomenon can be observed and the DNA-PAINT-based super-resolution imaging was realized. To be specific, first, SMLM processing was used to identify possible protein sites and image reconstruction was performed to obtain super-resolution images. Subsequently, the super-resolution image data were analyzed by qPAINT to obtain the expression level of the corresponding protein. qPAINT has been utilized in many articles [25,26,27]. Compared with existing quantitative methods, qPAINT can directly calculate the number of targets. By analyzing the predictable binding kinetics between the imager and docking strands, qPAINT counts the number of targets without spatially resolving them. qPAINT represents a conceptual framework that can simultaneously achieve high accuracy, precision, wide dynamic range, robustness, and multiplexing capability to quantify the number of labeled targets.

#### 3.2.1. Capture and Detection of Exosomes Secreted by Cells

In the experiment, the PD-L1 protein on exosomes was labeled by an antibody-DNA probe. A buffer containing imaging strands was introduced into the channel to realize fluorescence scintillation through the dissociation and binding of the docking strand and imaging strand. When the docking chain and the imaging chain are combined, the fluorescent signal can be captured, allowing the localization of the PD-L1 proteins. In the imaging process, the amount of liquid that can be retained in the microfluidic channel is tiny, hence, the concentration of the imaging chain in the imaging buffer was increased to obtain better scintillation images. Finally, 5000 frames of images were collected and the reconstructed super-resolution image is shown in Figure 6.

Figure 6A–C shows the SMLM image of exosome membranes, the DNA-PAINT image of PD-L1 protein on exosomes, and the merged image of these two channels, respectively. In the experiment, the location of exosomes was first determined by PKH26 membrane dye. Then, we switched to the Cy5 channel to achieve DNA-PAINT images of PD-L1 proteins. Cy5 was modified on the imaging chain, which was used for the luminescence of the imaging chain. It can be seen from Figure 6C that the fluorescence signals of the exosomal membrane channel and PD-L1 protein channel are highly coincident, indicating that we have successfully achieved DNA-PAINT images of exosomal PD-L1. As shown in Figure 6D, DNA-PAINT provides a high localization precision, which is beneficial for exosome imaging. From the cross-sectional fluorescence intensity profiles in Figure 6E, we can see that the FWHM (Full Width at Half Maximum) of the exosome derived from DNA-PAINT is 110 nm, while the FWHM in PKH26 channel under wide field imaging is 500 nm. The result of DNA-PAINT is much smaller than that of wide-field imaging and is in line with the actual size of exosomes. Therefore, compared with traditional wide-field fluorescence imaging, DNA-PAINT imaging can obtain superior spatial resolution and achieve more accurate exosome structure information.

For background fluorescence problems, we set an appropriate fluorescence intensity threshold in the process of fluorescence analysis to eliminate background fluorescence. After fixing the capture probes, bovine serum albumin (BSA) was added to block the spare sites to reduce nonspecific binding. Furthermore, the influence of non-specific binding of the aptamer with other substances was removed through dual-fluorescence co-localization of PKH26 and AF647. For a specific reaction site, the binding of exosome and PD-L1 aptamers would induce a perfect overlap of AF647 and PKH26 channels in the images. Otherwise, if only one kind of fluorescence signal is detected, such sites should be considered nonspecific reaction sites and eliminated.

#### 3.2.2. Quantitative Analysis of PD-L1 on the Surface of Exosomes Using qPAINT

In order to achieve an accurate quantitative analysis of PD-L1 on exosomes, qPAINT was utilized. According to the principle of qPAINT, the average dark state duration ($\tau_{d}^{*}$) of PD-L1 protein on exosomes and the inflow rate of a given imaging chain need to be obtained.

Therefore, the first step is to obtain the binding and dissociation parameters of a pair of docking and imaging strands. In the experiment, the DNA docking strand was directly modified on the surface of the chambered coverslip and subjected to qPAINT imaging. A total of 5000 frames of scintillation images were collected for the binding kinetics analysis of the imaging strand. Since the binding kinetic parameters of a single DNA strand were to be obtained, it was necessary to select the appropriate DNA strand concentration to ensure that only one DNA strand existed at each flickering site. After optimizing a series of concentrations, the final choice was to employ 300 μL 5 nM of the docking strand. Under these parameters, the docking strand in the chamber was evenly dispersed on the surface of the coverslip to eliminate the error caused by the agglomeration of multiple DNA strands.

During the imaging process, all parameters were kept consistent with the DNA-PAINT imaging of exosomal PD-L1. The super-resolution image reconstructed from 5000 frames of DNA strand scintillation images is shown in Figure 7B. It can be seen that the DNA docking strand was evenly distributed on the surface of the coverslip, and the obtained scintillation curve can be used to identify the binding and dissociation of a single docking strand.

A frame of image in the imaging process (Figure 7A) shows that transient fluorescent spots can appear in the whole field of view. For example, we could select two different areas in the image (Figure 7A), one is the location where the DNA docking strand exists, and the other is the blank area. Then, we could obtain their photoswitching curves. As shown in Figure 7C, for the spot in the red area, the fluorescence intensity curve occasionally showed strong fluorescence signals. While the fluorescence intensity in the green area was kept at a low level without obvious fluctuation. The reason is that the DNA docking strands bind and release the imaging strand randomly, resulting in a significantly higher Cy5 photoswitching frequency in the red region than in the background region.

After that, in order to obtain the expression level of exosomal PD-L1, qPAINT was used to analyze the data according to the method of Ralf Jungmann and his team [31].

In order to more accurately explore the effect of different concentrations of PD-L1 siRNA on the expression level of PD-L1 in tumor cells, cells were cultured with different concentrations of siRNA (20 nM, 30 nM, and 40 nM, and the blank control group). After continuous cell culture for 3~5 days, exosomes were captured in the channel and exosomal PD-L1 proteins were specifically labeled by antibody-DNA probe. Subsequently, 5000 frames of images in the Cy5 channel were collected. The raw data were imported into the Picasso software to calculate the corresponding $\tau_{d}^{*}$. After calibration ξ in the software, the content of PD-L1 proteins on exosomes was obtained.

The reconstructed DNA-PAINT image by Picasso software is shown in Figure 8A. The corresponding exosomal PD-L1 protein contents can be obtained by using Formula (2).

The density of exosomal PD-L1 corresponding to HeLa cells incubated with different concentrations of siRNA is shown in 8B. With the increase in siRNA concentration, the density of exosomal PD-L1 decreased, indicating that more siRNA strands have entered the cell to prevent the expression of PD-L1. Compared with the qualitative analysis based on fluorescence intensity, the quantification of PD-L1 on exosomes obtained by qPAINT is more reliable. Our research on this topic is consistent with previous reports [32,33,34,35].

## 4. Conclusions

In this study, the PD-L1 protein on the surface of tumor cell-derived exosomes was observed by DNA-PAINT. The PD-L1 protein was analyzed quantitatively through qPAINT and the inhibitory effect of PD-L1 siRNA on the expression level of exosomal PD-L1 protein was validated. As the concentration of siRNA increases, the density level of PD-L1 on exosomes decreases. When the concentration of PD-L1 siRNA was 40 nM, the inhibition efficiency of PD-L1 protein on the surface of HeLa cell exosomes reached 68.7%. Compared to traditional TIRF imaging, DNA-PAINT offers superior localization precision and spatial resolution. qPAINT was used to realize quantitative analysis of exosomal PD-L1 and the results confirmed that PD-L1 siRNA can effectively inhibit the expression level of PD-L1 proteins, providing new insights for immunotherapy. Furthermore, the microfluidic chip requires low sample quantities. In summary, the DNA-PAINT-based exosomal protein quantification chip enables convenient, rapid, and highly sensitive liquid biopsy, holding a good potential in early tumor detection and theranostics.

## Figures and Tables

**Figure 1 sensors-24-00173-f001:**
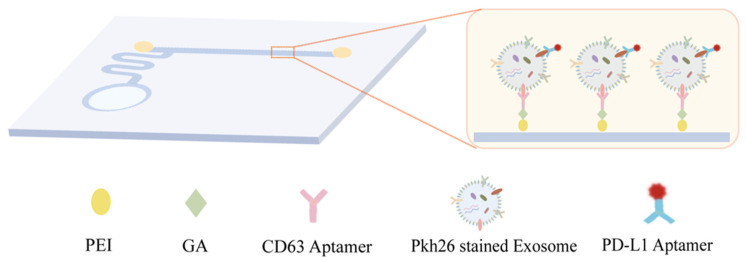
Schematic diagram and physical drawing of the microfluidic chip.

**Figure 2 sensors-24-00173-f002:**
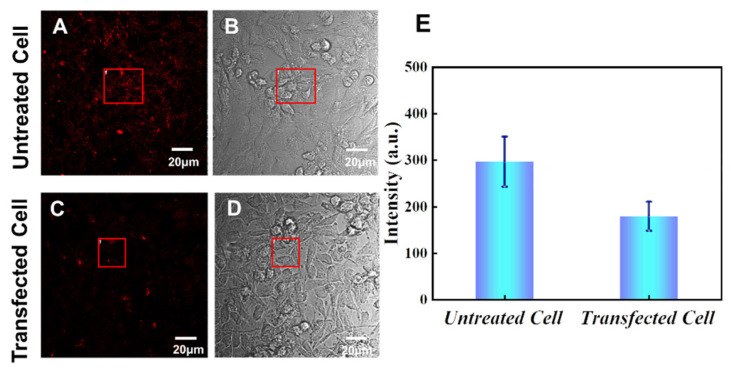
Immunofluorescence staining of PD-L1 proteins on the surface of HeLa cells and the corresponding fluorescence intensities. (**A**) Fluorescence image of untreated HeLa cells, (**B**) bright field image of untreated HeLa cells, (**C**) fluorescence image of transfected HeLa cells, (**D**) bright field image of transfected HeLa cells, (**E**) fluorescence intensities of PD-L1 on transfected and untreated HeLa cells. The intensities were obtained by the software integrated with the confocal microscope. The error bars are the standard deviations of 3 individual experiments. The part highlighted by the red border is for experimental fluorescence analysis.

**Figure 3 sensors-24-00173-f003:**
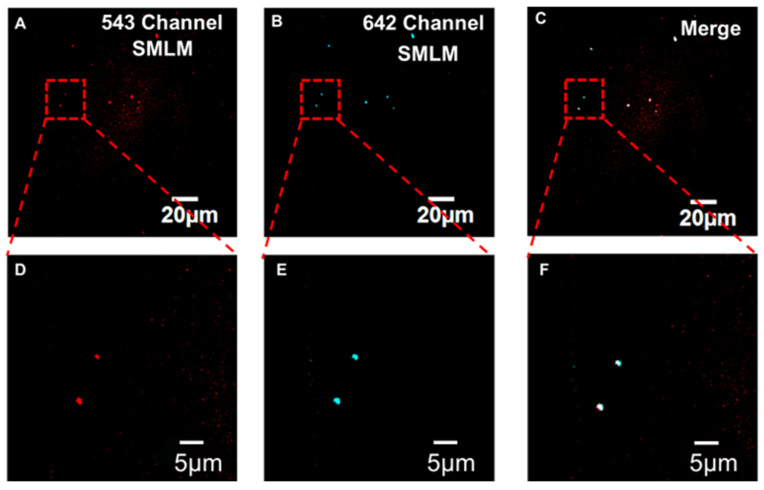
Fluorescence colocalization images of HeLa cell exosomes. (**A**) PKH26 membrane dye channel, (**B**) Alexa fluor 647@PD-L1 aptamer channel, (**C**) merged image of the two channels. (**D**–**F**) Enlarged view of the boxed areas in panels (**A**–**C**).

**Figure 4 sensors-24-00173-f004:**
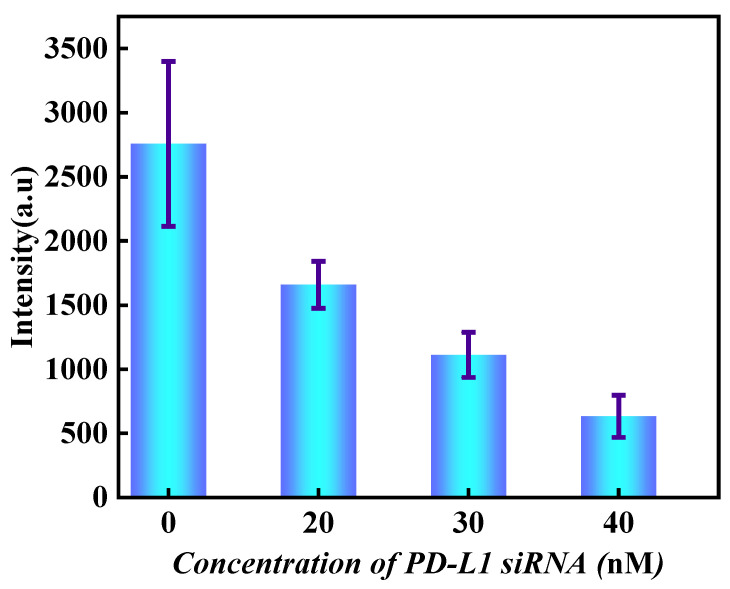
The down-regulation effect of different concentrations of PD-L1 siRNA on the expression levels of PD-L1 on HeLa exosomes. The error bars are the standard deviations of 15 individual experiments.

**Figure 5 sensors-24-00173-f005:**
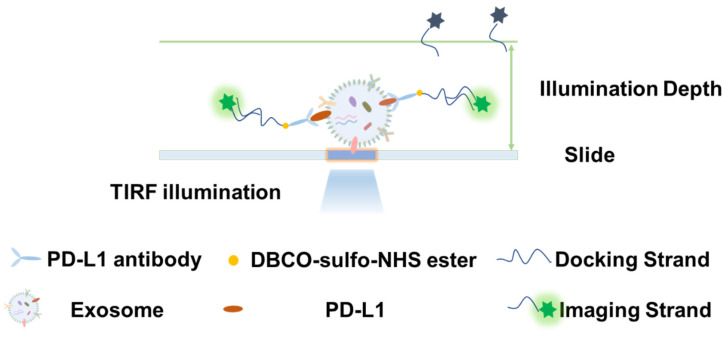
Schematic illustration of the detection mechanism of exosomal PD-L1 using DNA-PAINT.

**Figure 6 sensors-24-00173-f006:**
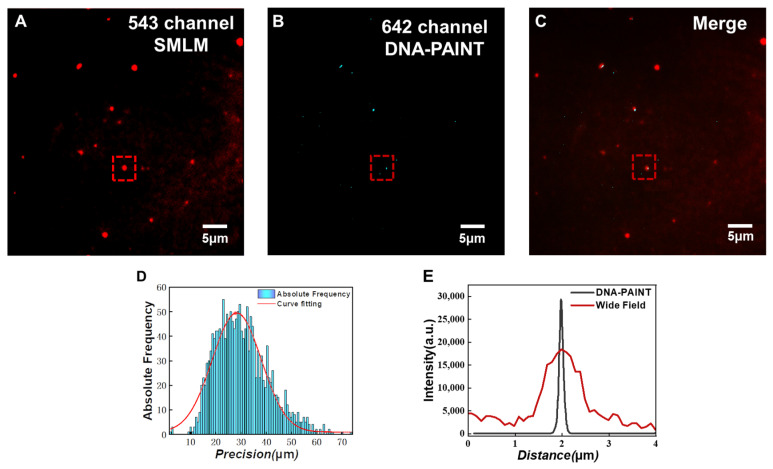
Super-resolution images of HeLa cell exosomes. (**A**) SMLM image of the PKH26 exosome membrane dye channel, (**B**) DNA-PAINT image of PD-L1 protein on the surface of exosomes, (**C**) The merged image of (**A**,**B**). (**D**) The localization accuracy of DNA-PAINT image. (**E**) Fluorescence intensity curve of the exosome cross section as indicated by the red square in (**B**).

**Figure 7 sensors-24-00173-f007:**
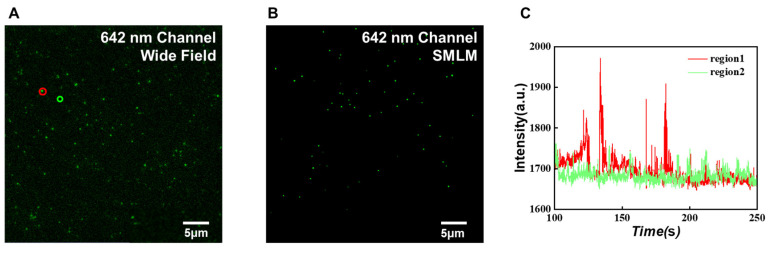
DNA single strand quantitative analysis. (**A**) A wide field image of the DNA strand fixed at the bottom of the chamber coverslip. (**B**) DNA-PAINT image of the DNA strand fixed at the bottom of the chamber coverslip. (**C**) A fluorescence photoswitching curve of the marked areas in (**A**).

**Figure 8 sensors-24-00173-f008:**
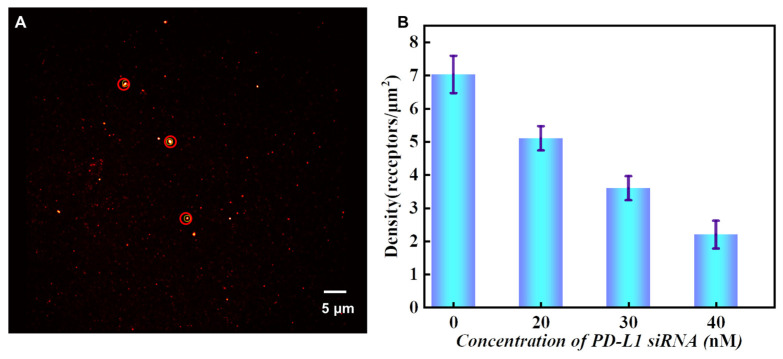
Quantitative analysis of exosomal PD-L1. (**A**) A DNA-PAINT image of exosomal PD-L1 produced by the Picasso software. (**B**) The density of exosomal PD-L1 corresponding to HeLa cells treated by different concentrations of PD-L1 siRNA. The error bars are the standard deviations of 15 individual experiments. The highlighted part of the red circle is used for experimental fluorescence analysis.

## Data Availability

The datasets used or analyzed during the current study are available from the corresponding author on reasonable request.

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
