# Peer review of "Fluorescence Super-Resolution Imaging Chip for Gene Silencing Exosomes"

_sensors, 2023, doi:10.3390/s24010173_

Round 1

Reviewer 1 Report

Comments and Suggestions for Authors

The article is devoted to the study of the breast cancer biomarker PD-L1 on the surface of exosomes secreted by tumor cells using the DNA-PAINT method. The authors propose to use a microchip that integrates cell culture and exosome capture, as well as specific aptamers for labeling PD-L1. The authors also investigate the effect of small interfering RNAs (siRNAs) on the level of expression of PD-L1 in cells and exosomes. The article is of interest for modern biomedicine, as exosomes are potential targets for cancer immunotherapy, and PD-L1 is one of the key immune checkpoints that regulate the interaction of tumor and immune cells. However, the article has a number of drawbacks and questions that need to be eliminated or clarified before publication. Below are the comments and questions on the article: Abstract: it would be desirable to see here a phrase, how the contribution of the authors differs from the existing works on the topic and what advantages the proposed approach has.

After the introduction, it would be desirable to see a section with materials and methods. Why is it located after the Conclusion? In the main part, the results of statistical analysis of data, such as significance level, confidence intervals, correlation and regression coefficients, are not provided. The authors should justify the choice of statistical methods and show that the obtained results are not random or erroneous.

In section 2.1. only the scheme of the chip is given. Is it possible to provide a photo of the manufactured chip in the assembly? Why was this design of the chip chosen?
In section 2.1.2. the authors did not explain why they used CD63 aptamer for capturing exosomes, and not other biomarkers, such as CD9 or CD81. What is the specificity and sensitivity of CD63 aptamer to exosomes? How did they control the background fluorescence and nonspecific binding of the aptamer with other substances in the culture medium?

In section 2.2. the authors did not provide evidence that PD-L1 antibody and PD-L1 aptamer actually bind to PD-L1 on the surface of exosomes. How did they exclude the possibility of cross-reaction with other molecules on exosomes or in solution? How did they confirm the specificity and affinity of their probes to PD-L1?

In section 2.2.1. the authors did not describe how they determined the localization of exosomes using two-color fluorescent colocalization of PKH26 and AF647. What algorithm or software did they use for this? How did they account for possible diffusion or detachment of probes from exosomes during measurement?
In section 3. the authors did not discuss how their results agree or contradict with other studies on the topic. What is the biological relevance of their findings? What are the possible mechanisms explaining the effect of siRNA on the level of PD-L1 in cells and exosomes? In the conclusion, no practical recommendations are given that would follow from the results of the study. The authors should indicate how their work can be used for applied purposes, such as diagnosis, treatment, prevention or control of breast cancer.
Thus, after minor revision and elimination of the described comments, the article can be recommended for publication.

Comments on the Quality of English Language

The article requires a minor revision of the English language.

Reviewer 2 Report

Comments and Suggestions for Authors

This manuscript develops a useful microfluidic chip by combining cell cultivation and exosomes detection modules to study the gene silencing effect of PD-L1 siRNA. DNA-PAINT technology is employed to investigate the spatial localization and quantitative analysis of PD-L1 protein on the surfaces of HeLa cells and MDA-MB-231 breast cancer cells. This work could provide an efficient integrated platform for the study of PD-L1 related tumor immunotherapy and have a good promising potential for practical applications. Therefore, I recommend that this article can be published after a major revision.

1. According to the caption of Figure 1, Figure 1 lacks the actual photograph of the microfluidic device.

2. The illustration in the manuscript at line 112-114 should be added with literatures.

3. Calculation method of fluorescence intensities should be introduced in the manuscript.

4. The results of investigations should be summarized in the conclusion of the manuscript.

5. Wrong illustrations should be corrected at the line 60, 182, 217, 236-238.

6. Description of the Figure 5 should be clearer and more organized.

7. English language needs to be improved.

Comments on the Quality of English Language

Moderate editing of English language required

Round 2

Reviewer 2 Report

Comments and Suggestions for Authors

I think this work is publishable in Sensors.